# Assessing Forest Level Response to the Death of a Dominant Tree within a Premontane Tropical Rainforest

**Manuel R. Flores III** [1,2]**, Luiza Maria Teophilo Aparecido** [3]**, Gretchen R. Miller** [4] **and Georgianne W. Moore** [1,*]

1. Department of Ecology and Conservation Biology, Texas A&M University, College Station, TX 77840, USA; manuelf3@illinois.edu
2. Department of Plant Biology, University of Illinois Urbana-Champaign, Champaign, IL 61801, USA
3. School of Earth and Space Exploration, Arizona State University, Tempe, AZ 85281, USA; lmtaparecido@gmail.com
4. Department of Civil and Environmental Engineering, Texas A&M University, College Station, TX 77840, USA; gmiller@civil.tamu.edu
* Correspondence: gwmoore@tamu.edu; Tel.: +1-979-845-3765

**Abstract:** Small-scale treefall gaps are among the most important forms of forest disturbance in tropical forests. These gaps expose surrounding trees to more light, promoting rapid growth of understory plants. However, the effects of such small-scale disturbances on the distribution of plant water use across tree canopy levels are less known. To address this, we explored plant transpiration response to the death of a large emergent tree, *Mortoniodendron anisophyllum* Standl. & Steyerm (DBH > 220 cm; height ~40 m). Three suppressed, four mid-story, and two subdominant trees were selected within a 50 × 44 m premontane tropical forest plot at the Texas A&M Soltis Center for Research and Education located in Costa Rica. We compared water use rates of the selected trees before (2015) and after (2019) the tree gap using thermal dissipation sap flow sensors. Hemispherical photography indicated a 40% increase in gap fraction as a result of changes in canopy structure after the treefall gap. Micrometeorological differences (e.g., air temperature, relative humidity, and vapor pressure deficit (VPD)) could not explain the observed trends. Rather, light penetration, as measured by sensors within the canopy, increased significantly in 2019. One year after the tree fell, the water usage of trees across all canopy levels increased modestly (15%). Moreover, average water usage by understory trees increased by 36%, possibly as a result of the treefall gap, exceeding even that of overstory trees. These observations suggest the possible reallocation of water use between overstory and understory trees in response to the emergent tree death. With increasing global temperatures and shifting rainfall patterns increasing the likelihood of tree mortality in tropical forests, there is a greater need to enhance our understanding of treefall disturbances that have the potential to redistribute resources within forests.

**Keywords:** treefall gap; sap flux; plant physiology; tropical ecohydrology

## 1. Introduction

Treefall gaps occur in all forest types and classically represent mild disturbances that influence both ecosystem succession, biodiversity, and the coexistence of species [1–3]. This complex process has led to an influx of studies assessing tree gap dynamics in order to understand their ecological significance. Perhaps the most prominent contribution of treefall gaps is their impact on species diversity. From temperate forests residing in the United States to the tropical forests of South America, treefall gaps allow for a release of canopy-piercing light, promoting understory growth and resulting in increased species diversity as well as heterogeneity within the forest structure [1,4–6]. Many studies have assessed this phenomenon in the attempt to empirically understand the true effects of such disturbances. Studies such as Lajtha [7] have worked to quantify how the release of nutrients from treefall gaps influence the growth of various species that differ in their

nutrient uptake ability. Meanwhile, studies such as Uhl, et al. [8] have conducted a more exhaustive analysis to explore seedling recruitment and growth rate of various species spanning multiple functional groups and placement in forest understory. Studies of this nature highlight not only the heterogeneity of gap formation within environments but also the differences in ecological significance that occur between them as well [2,3].

Despite widespread interest in "gap dynamics" [1], there have been markedly fewer publications assessing how water use among neighboring trees is altered from treefall gaps in natural systems. The lack of academic focus on this topic may have arisen as a consequence of the assumption that the tree thinning is a prerequisite to observing substantial increases in tree water use and availability [9–12] or is possibly due to the fact that there are soil and climatic factors [13,14] that may make assessing the hydrological significance of these events more challenging, especially in natural forest systems. Literature on treefall gaps is particularly lacking in tropical forests. Here, water use is scarcely analyzed in conjunction with treefall gaps, possibly due to limitations in both identifying and dating the tree gaps within dense tropical forests and in the ability to assess tree water use both pre- and post-canopy gap formation. However, rainfall has been shown to significantly influence species turnover in conjunction with soil phosphorus concentration, highlighting the need to better understand water use dynamics within highly biodiverse tropical forests [15]. Furthermore, understanding how tropical treefall gaps influence forest level water use is pertinent due to the variability of plant individuals within a highly biodiverse tropical forest, especially concerning early life stage water-use efficiency [16]. Partitioning the influence of treefall gaps on water usage among species residing at different canopy levels provides additional insight as to how this disturbance and the following increase of light is distributed among species at differing canopy levels.

To investigate the influence of a treefall gap on the water use of remaining trees, this study analyzed the change in sap flux ($Js$) rates of suppressed, mid-story, and dominant trees after the death of a large emergent tree, *Mortoniodendron anisophyllum* (Standl.) Standl. & Steyerm, in an experimental forest plot located in a tropical premontane forest in Costa Rica. The death of this species occurred as a consequence of an internal stem fungal and pathogen infection first observed in December of 2018, and its imminent collapse occurred in November of 2019 (Figure 1). The death of this emergent tree provided a unique opportunity for treefall gap analysis due to its ecological significance in the experimental plot it resided in. Prior to its death, this tree was the largest tree recorded in the area, towering over its neighbors with an approximate height of 40 m and diameter-at-breast-height (DBH) greater than 220 cm [17]. Additionally, this tree's size made its water use extremely high and was once estimated to be responsible for 36% of the average water use (measured as sap flux) recorded in the plot, as well as championed as the highest water user in the world according to SAPFLUXNET [17–19]. From this information, it is not unreasonable to hypothesize that the death of this tree would have profound ecological effects on the forest surrounding it.

To this end, this study used thermal dissipation sensors to measure water use among trees neighboring the dead *M. anisophyllum* tree, in order to specifically investigate (i) how total water use was altered for all measured trees pre- and post-emergent tree death and (ii) how water use differed between overstory and understory trees both pre- and post-emergent tree death in an effort to illustrate the significance that treefall gaps may have on forest level water relations and altering stand microclimate. This work built upon the existing experimental infrastructure documented in Aparecido, Miller, Cahill and Moore [17], whose work provided sap flux measurements of sample trees pre-tree death, allowing for this study to capture comparable water use dynamics among the neighboring trees of the *M. anisophyllum*.

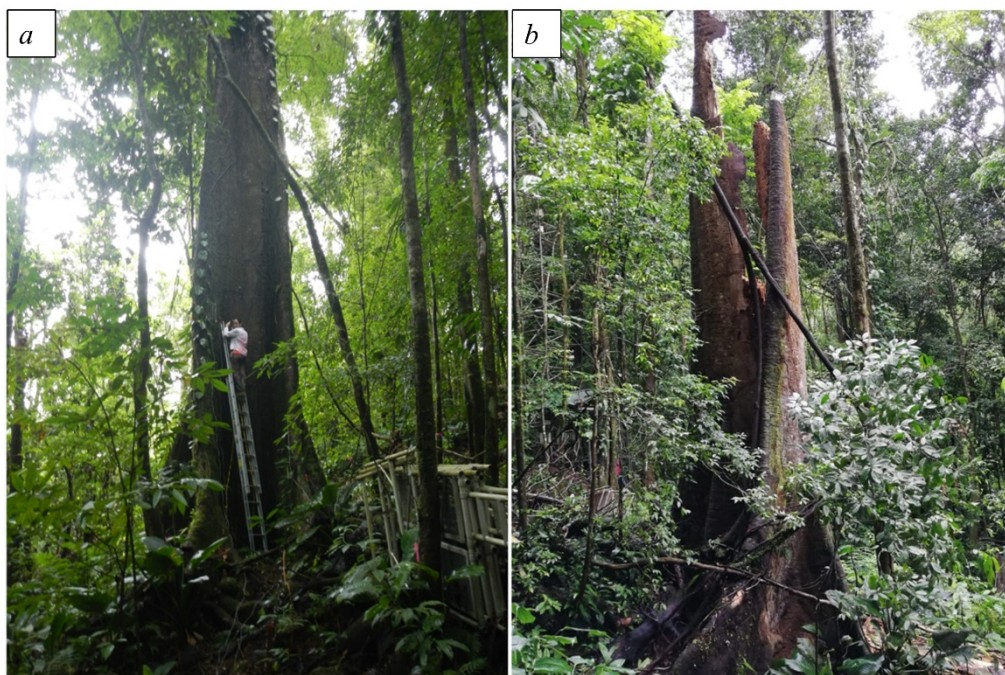

**Figure 1.** Image of the emergent *M. anisophyllum* species pre-death (**a**) and post-death (**b**). Photo credits: Luiza Aparecido (**a**); Eugenio Gonzalez (**b**).

## 2. Materials and Methods

### 2.1. Study Site

This study was conducted in a mature transitional wet tropical forest at the Texas A&M Soltis Center for Research and Education within the Alajuela Province of Costa Rica (Figure 2). This site is located at 10°23′13″ N–84°37′33″ W and exists in close proximity to San Isidro de Peñas Blancas at an elevation of ~600 m. The area receives ~4200 mm of precipitation annually and has an average annual air temperature of ~24 °C [20]. There is a distinct wet season occurring between the months of May and December, with an average monthly rainfall of ~470 mm and a dry season in which precipitation is still present but markedly lower, ~195 mm per month [17,20]. The experimental research plot resides on a 45° sloped terrain, has an area of 2200 m$^2$, and is centered around a 42 m tall micrometeorological research tower. There are 56 botanically identified tree species located in this plot, with 151 individuals present. Of these species, *Carapa guianensis* Aubl. has the highest number of individuals present (31 individuals), while *M. anisophyllum* species are the largest trees found in the plot and make up a disproportionate amount of the basal area present (i.e., all three *M. anisophyllum* trees account for 54% of plot basal area). Tree size within this plot is highly variable, with DBH ranging from 6 to 220 cm and height ranging from 6 to 40 m. Canopy structure consists of interwoven tree crowns between individuals, with frequent small canopy gaps distributed throughout the plot. Tree dominance classification (i.e., subdominant, mid-story, suppressed), height estimates, and species plot data were used from field observations collected by Aparecido, Miller, Cahill and Moore [17]. Previous studies have shown the importance of this canopy structure for accurately representing the forest in models of the land surface and its carbon and water cycling [21].

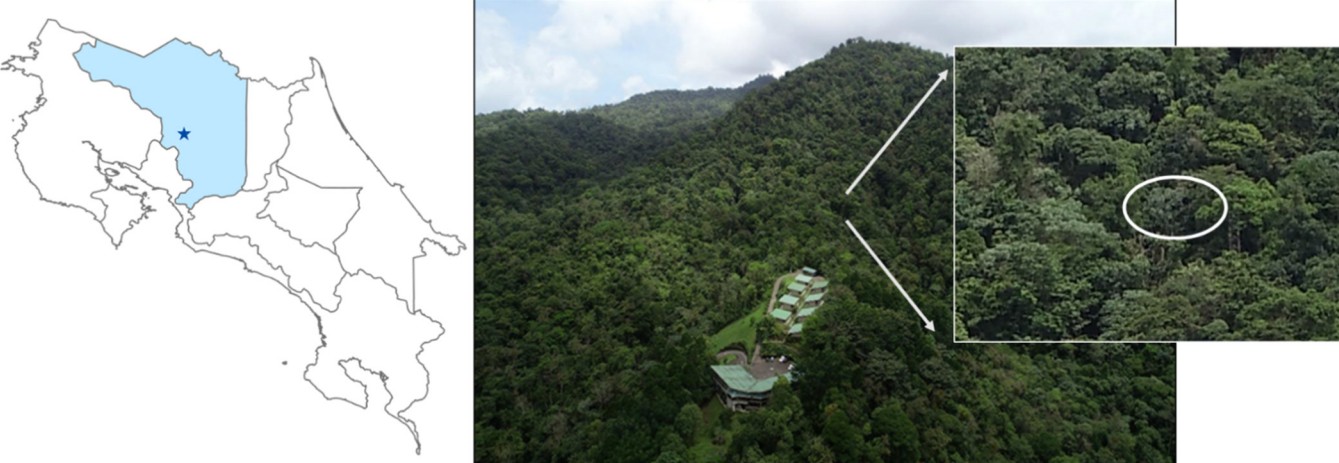

**Figure 2.** Map of Costa Rica with the Soltis Center's location along with an aerial UAV photo of the Soltis Center and the surrounding forest that marks the site the *M. anisophyllum* before its death (circled in inset). Photo credits: Elizabeth M. Prior and Kelly Brumbelow.

### 2.2. Sap Flux Measurements

The collection of 2019 (post-tree death) water use measurements followed the same protocols used in Aparecido, Miller, Cahill and Moore [17] and consisted of continuous diurnal measurements of sap flux density ($J_s$, kg m$^{-2}$ h$^{-1}$) using 12 heat dissipation sensors [22] across 3 suppressed, 4 mid-story, and 2 subdominant trees (Table 1). The number of sensors varied by tree size, with suppressed trees receiving one sensor per tree, while both mid-story and subdominant trees received two sensors when possible. However, due to sensor failure and time constraints, not all subdominant and mid-story trees received multiple sensors. For trees with multiple sensors, the mean value for both sensors was used as the tree's $J_s$. Sensors were constructed as detailed in Phillips, et al. [23] and installed in the outermost xylem of our sample trees over a 5-week period between the months of June and July in 2019. All sensors were installed at breast height (1.30 m) or above tree buttresses, if present, and each set of sensors contained probes 2 cm in length (as recommended by Clearwater, et al. [24]). The 2015 sensors were preferentially placed on side slopes when accessible, and 2019 sensors were installed in areas clear of previous sensor wounds and placed opportunistically due to terrain. Wounding effects were minimized by new installations in 2019. Following Aparecido, Miller, Cahill and Moore [17], radial changes in sap flux density were not taken into account. Sap flux data were collected every 30 s and averaged as 10 min intervals using a CR1000 datalogger (Campbell Scientific Inc., Logan, UT, USA). Sap flux measurements used in this study were recorded over a 183-day period between 2 July 2019 and 31 December 2019 and were compared with the same 183-day date span of 2015 (pre-tree death) that was collected by Aparecido, Miller, Cahill and Moore [17]. All trees measured in 2019 were trees measured in 2015 by Aparecido, Miller, Cahill and Moore [17] using their identical methodology.

**Table 1.** Plant descriptions of the three suppressed, four mid-story, and two subdominant trees sampled. Tree measurements taken from Aparecido, Miller, Cahill and Moore [17].

| Canopy Level | Tree Species | Number of TD Sensors | Diameter at Breast Height (DBH cm) | Height (m) | Basal Area (m²) | Sapwood Area (m²) |
|---|---|---|---|---|---|---|
| Subdominant | *Otoba novogranatensis* | 1 | 62.8 | 29 | 0.31 | 0.192 |
| Subdominant | *Genipa americana* | 2 | 46.2 | 28 | 0.168 | 0.102 |
| Mid-story | *Ampelocera macrocarpa* | 1 | 15.6 | 16 | 0.019 | 0.012 |
| Mid-story | *Carapa guianensis* | 1 | 17.3 | 16 | 0.024 | 0.016 |
| Mid-story | *Ampelocera macrocarpa* | 2 | 32 | 26 | 0.08 | 0.066 |
| Mid-Story | *Eschweilera* sp. | 2 | 30.5 | 27 | 0.073 | 0.053 |
| Suppressed | *Trophis mexicana* | 1 | 10 | 11 | 0.008 | 0.006 |
| Suppressed | *Carapa guianensis* | 1 | 8.3 | 9 | 0.005 | 0.004 |
| Suppressed | *Cupania macrophylla* | 1 | 6.9 | 10 | 0.004 | 0.003 |

Each sap flux sensor consisted of a reference and heater probe that measured temperature differences (mV). By using an empirical calibration equation developed by Granier (1987), (Equation (1)):

$$J_s = 0.119 \left( \frac{\Delta T_M - \Delta T}{\Delta T} \right)^{1.231} = 0.119 K^{1.231}, \tag{1}$$

Sap flux density ($J_s$) was calculated with maximum temperature differences when sap flux is assumed to be 0 ($\Delta T_M$), as well as the actual temperature difference at the time of measurement ($\Delta T$) [17]. After this, total tree $J_s$ was calculated for each tree and, as mentioned earlier, total tree $J_s$ for trees housing multiple sensors was determined by averaging the number of sensors installed to represent the tree's average $J_s$ rate across a large and possibly non-uniform sapwood area. Sap flux density was then consolidated into hourly totals (kg m$^{-2}$ h$^{-1}$) among individual trees that had their water use compared between 2015 and 2019.

The data were screened for erroneous values due to sensor failure or out-of-range voltage readings, which were indicative of power loss, insect damage, or other environmental factors. The 2019 omitted data set (~6.5% of values) was then gap-filled using linear regression methods as well as interpolation. For more extended gaps and when possible, the response at a given sensor was determined by using its correlation with other sensors, but only when the relationship between the two exceeded $R^2 > 0.6$. Lastly, when power was lost for all sensors and interpolation/extrapolation could not be employed, measurements were omitted for all sensors (<0.5% of all 2019 measurements).

### 2.3. Assessing Forest Change

To assess any change in forest structure as a result of emergent tree death, gap fraction measurements were conducted at 9 individual points in 2019 within the plot (12.5 × 12.5 m grid) and were compared with 2014 and 2016 gap fractions separately. For all years, high resolution hemispherical photography was collected using a digital camera (D90, Nikon, Melville, NY, USA) equipped with an 8 mm fisheye lens (F3.5-HD, Rokinon, New York, NY, USA). All images were analyzed using HemiView software version 2.1 (Delta-T devices Ltd., Cambridge, UK), with gap fraction analysis conducted under a threshold of 217. While gap fraction measurements were collected in 2015, differences in the camera and lens used to collect these data resulted in incomparable values, and thus they were not included in this analysis. Data were reanalyzed for 2014 and 2016 under the site characteristics used in the 2019 analysis, with the only differences in analysis consisting of the magnetic declination assigned for the site for each separate year. Additionally, photographs taken in 2019 were the only photos taken with a North orientation marker. However, purposefully orienting the photographs once imported into HemiView was found to have little to no effect on the calculated gap fraction, and thus all images were simply imported and analyzed without additional orientation.

### 2.4. Micrometeorological Data

Measurements were collected at a height of 3 m above the ground surface from a meteorological tower (MET) located on the forest edge of the Soltis Center. Twelve months of data were collected at 5 min intervals for relative humidity (*RH*, %, HMP50, Campbell Scientific, Logan, UT, USA) and air temperature (*T*, °C, temperature probes model 107, Campbell Scientific, Logan, UT, USA)) for 2015 and 2019. These micrometeorological data were collected in order to evaluate the possibility of climate variability (e.g., drought) affecting water uptake of the trees sampled pre- and post-tree death of the *M. anisophyllum*, thus not being related to the effects of a forest gap. From these measurements, vapor pressure deficit (*VPD*) was calculated using the following equations (Equations (2) and (3)):

$$e^o = 610.7 \times 10^{7.5T/(237.3+T)}, \tag{2}$$

$$VPD = e^o \times \left(1 - \frac{RH}{100}\right) \times 1000, \tag{3}$$

where, saturation vapor pressure ($e^o$) is first calculated using temperature in Celsius (*T*) and multiplied by the remaining moisture free atmosphere to calculate *VPD* (kPa). Precipitation measurements were collected daily at the base of the MET tower using a manual graduated rain gauge.

Additionally, photosynthetically active radiation (PAR, µmol m$^{-2}$ s$^{-1}$) data within the vicinity of the sap flow trees were also collected on a canopy access tower centered in the research plot. PAR data were recorded at 30 min intervals at heights of 10 m, 21 m, 27 m, and 38 m using quantum sensors (LI-191, LI-COR, Lincoln, NE, USA). Monthly average PAR data from 2017, rather than 2015, were compared with 2019 due to sensor failure in 2015.

### 2.5. Data Processing/Analysis

A two-tailed two-sample *t*-test assuming equal variance ($\alpha = 0.05$) was conducted to evaluate differences in understory, overstory, and total water use pre- and post-dominant tree death. A similar test was also carried out to compare differences in water usage across canopy levels within 2015 and 2019, respectively. To account for a low sample size of the subdominant size category, data from mid-story and subdominant trees were combined and considered as the overstory group (O), while suppressed trees were redefined as the understory (U) for the aforementioned analysis [17]. Additionally, two separate two-tailed two-sample *t*-tests assuming equal variance were conducted for gap fraction measurements of 2014 and 2016 against 2019 ($\alpha = 0.05$).

Daily averages of *RH*, *T*, and *VPD* MET tower data spanning the same days as our sap flux measurements were compared as yearly averages using a two-tailed two sample *t*-test assuming equal variance. Days where all or most data were missing for either year were omitted from analysis (n > 95 days for all MET parameters). Annual rain gauge data were analyzed similarly; however, no data filtering was necessary, and monthly measurements were compared instead of daily averages (n = 6). Additionally, daily averages of vertical PAR profiles were also analyzed through linear regression analysis in R Studio, version 4.0.2 [25,26] using the "car" package [27], and averages of the entire year were analyzed (n > 250 days for each sensor height). This was done in conjunction with a two-tailed two sample *t*-test assuming equal variance in order to better observe the effect of year (pre- and post-tree death), sensor height, and the interaction between year and sensor height (i.e., how the effect of sensor height may be different pre- and post-tree death) under a type 3 sums of squares analysis (regression = logPAR ~ Year + Height + (Year × Height)). PAR data were log-transformed, and sensor height was interpreted as integers rounded to the nearest meter prior to regression analysis. Lastly, November–December of 2019 PAR data were nearly completely missing for all sensor heights in 2019 (Figure 3a), resulting in no daily comparisons being done for data within these months for either year.

## 3. Results

### 3.1. Micrometeorological Data

We found significant differences between years in PAR (2017 = 53.25 ± 1.49 (SE) μmol m$^{-2}$ s$^{-1}$, 2019 = 74.60 ± 1.85 μmol m$^{-2}$ s$^{-1}$, $p < 0.0001$) and significant effects of year and light sensor height within our linear regression analysis ($p_{Year} < 0.0001$ and $p_{Height} < 0.0001$). However, there was no significant effect for the interaction term Year x Sensor Height for PAR ($p = 0.63$, Figure 3a). Annual micrometeorological comparisons showed that air temperature was ~2 °C higher in 2015 (24.77 ± 0.115 °C) than in 2019 (22.70 ± 0.135 °C, $p < 0.0001$, Figure 3b). *VPD* averaged lower in 2015 (0.205 ± 0.0162 kPa) than in 2019 (0.394 ± 0.0176 kPa, $p < 0.0001$, Figure 3c). *RH* was also higher by ~8% in 2015 (94.28 ± 0.48%) than 2019 (86.82 ± 0.71%, $p < 0.0001$, Figure 3d). Lastly, differences in precipitation were found to be statistically insignificant between both 2015 (494.65 ± 58.70 mm) and 2019 (412.17 ± 20.70 mm, $p = 0.21$, Figure 3e).

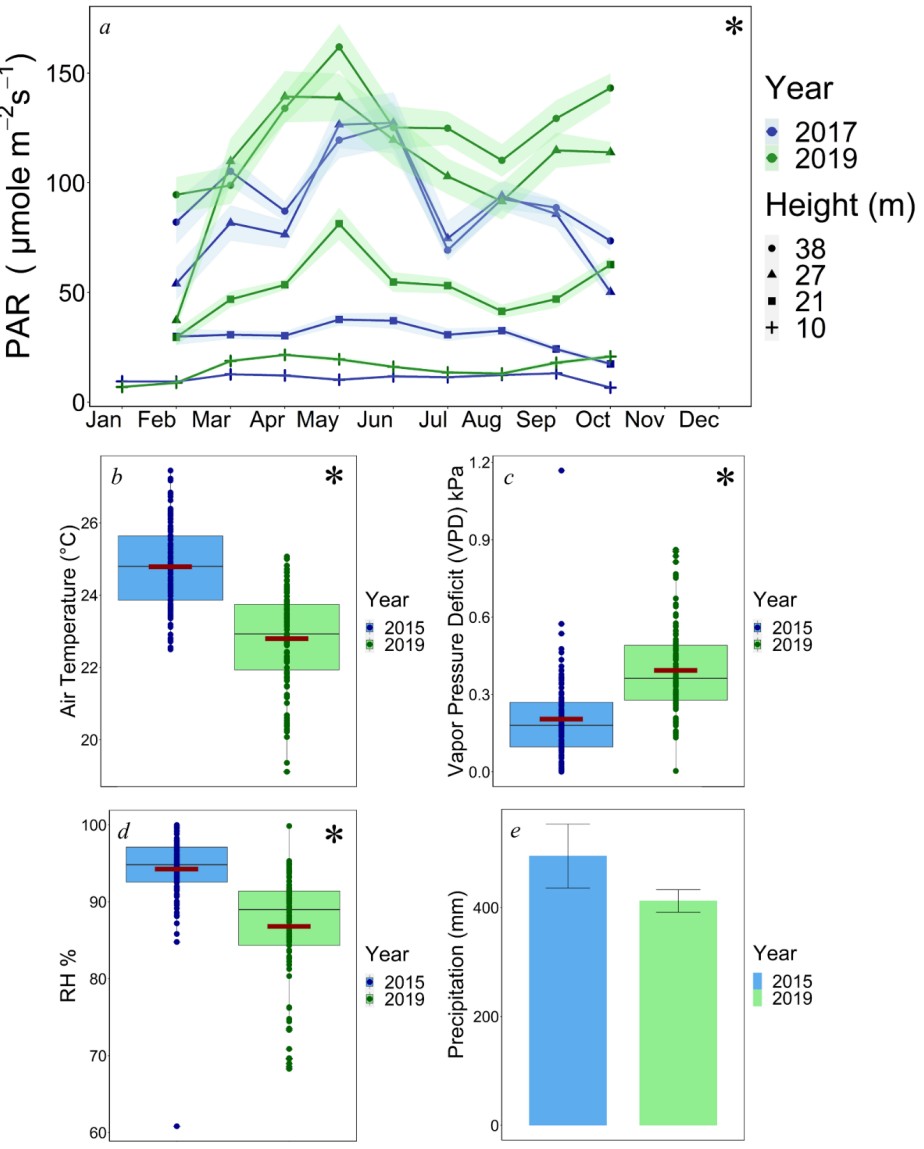

**Figure 3.** (**a**) Graph of monthly PAR averages for each sensor height for 2017 and 2019. (**b–d**) Side-by-side boxplots of air temperature, vapor pressure deficit, and relative humidity for 2015 and 2019. (**e**) Precipitation bar graph of 2015 and 2019. Light colored bands and error bars denote standard errors (**a,e**). Maroon bars indicate means, and asterisks denote significant differences (**b–d**).

### 3.2. Water Usage Comparisons

Sap flux of all trees averaged 15% higher in 2019 (16.06 ± 1.84 (SE) kg m$^{-2}$ h$^{-1}$) than 2015 (13.96 ± 1.70 kg m$^{-2}$ h$^{-1}$) when comparing measurements of the same trees. However, this difference was not statistically significant due to large tree-to-tree variation, as is common in sap flow studies ($p = 0.42$, Figure 4a,c). Prior to tree death, the overstory used an average of 21.73% more water in 2015 (14.94 ± 1.53 kg m$^{-2}$ h$^{-1}$) than the understory (12.01 ± 4.49 kg m$^{-2}$ h$^{-1}$, $p = 0.45$), again with high tree-to-tree variation (Figure 4a,c). Water use also did not differ between canopy layers in 2019; however, understory water usage (16.28 ± 4.43 kg m$^{-2}$ h$^{-1}$) was on average 2% *higher* than overstory water usage (15.95 ± 2.05 kg m$^{-2}$ h$^{-1}$, $p = 0.94$, Figure 4a). Overstory and understory tree water use was similar among years ($p_{Overstory} = 0.7$, $p_{Understory} = 0.54$). However, understory trees exhibited a 35.51% increase in water use compared with only 6.73% increase by overstory trees among years (Figure 4a,c).

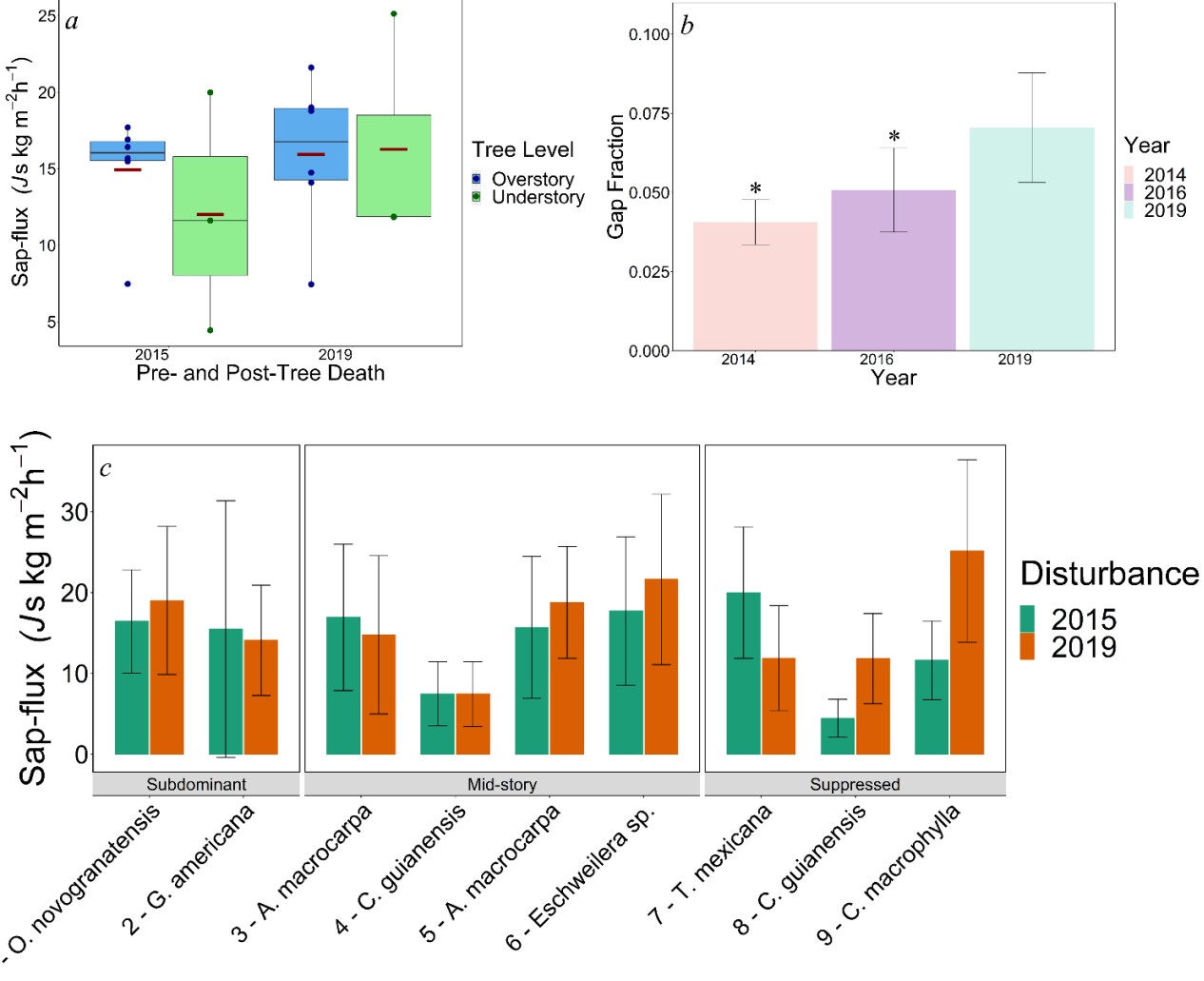

**Figure 4.** (**a**) Graph of average sap flux (*Js*) of overstory (O) and understory (U) trees sampled in 2015 and 2019. Each maroon marker indicates the mean for its respective canopy level. (**b**) Average gap fraction for 2014, 2016, and 2019. (**c**) Average individual tree sap flux from 2015 and 2019 study periods. Numbers in figure (**c**) identify species with respect to Table 1. Error bars on figures (**b**,**c**) indicate standard deviations. Asterisks indicate statistical significance.

### 3.3. Gap Fraction

Gap fraction was significantly lower in 2014 (0.04 ± 0.002) and 2016 (0.05 ± 0.004) when compared with 2019 post-tree death (0.07 ± 0.006, $p_{2014}$ = 0.0001, $p_{2016}$ = 0.01, Figure 4b).

## 4. Discussion

We hypothesized that a single treefall gap results in more light penetrating deeper into the canopy, which is expected to increase photosynthesis and air circulation that dries out trees and promotes higher water usage of the understory plants. Although we found high tree-to-tree variability for the nine individuals analyzed in this study and no statistically significant response in tree water use between pre- and post-emergent tree death, average understory tree water use trended strongly upward after the gap formed, surpassing that of overstory trees. Light gap fraction was clearly higher, and while this measurement may have also been affected by other treefalls in the area, PAR penetrated lower in the canopy after the tree fell. The effect may have been diminished during the wet season when the majority of our data were collected. More sunlight in the drier months could have enhanced water use by understory trees. On the contrary, more air circulation in the canopy gap to dry out wet leaves in the wet season could be a critical driver of observed differences. The size of the treefall gap may have also been a factor that we could not account for in this study. Small but significant differences in weather between years could possibly explain the differences observed, especially when considering the higher *VPD* means present in 2019 measurements. Nonetheless, the overstory had previously been responsible for ~22% more water consumption than understory individuals pre-tree death; the shift to overstory trees using ~2% less water than understory trees suggests a growth response from the forest understory after the gap was formed.

Our study found no evidence that the gap induced more stressful conditions where the emergent canopy partially shaded and protected overstory canopies. Rather, the event of the *M. anisophyllum* death likely enabled more light to pierce the canopy, allowing for understory trees to take advantage, possibly not only of increased water availability but also of increased light, explaining the relatively greater water use of understory trees after emergent tree death. Therefore, we expect reduced light competition rather than reduced water competition to drive plant responses to treefall gaps in this region.

It should also be noted that this study took place almost immediately following this large gap formation. Gap dynamics have been found to change as regrowth occurs, in terms of shifting light regimes and species' competitive response [28]. Eventually we would expect the forest plot to return to previous water use conditions as the forest canopy starts closing with new trees expanding their canopies. As new growth fills the gap and PAR decreases with canopy closure, we suspect water usage will re-equilibrate with overstory trees dominating in terms of water use. However, if stressful environmental conditions, such as droughts or sparser, more intense rainfall events increase in this region, the forest stand could see an increase in emergent tree mortality (i.e., hydraulic failure or physical damage from storms [29–31]), which could lead to a greater shift in species composition and water usage. The fact that the emergent *M. anisophyllum* tree used such a drastically different amount of water compared with its neighboring trees makes it unclear how this significant release of light and water will be distributed across the surrounding forest as gap-colonizing species germinate and begin to compete for resources. Given the high frequency of such small-scale disturbances throughout tropical forests, this study suggests that some amount of water use redistribution across canopy layers is likely, but it may take a very large gap or numerous mortality events of emergent trees coupled with more drastic climate shifts to yield a significant response.

Another potential contributor to high tree-to-tree variability obscuring our comparisons before and after the tree fell is the individual functional traits among the different species tested (such as early pioneer species and late successional species), as well as their placement within the forest plot along the steep east-facing slope. Despite the physiological

diversity in our plot, current evidence suggests that overall tropical tree water use may be less related to species-specificity and more directly dictated by environmental factors, plant size/architecture, and placement within a canopy [32–34]. Studies such as those of O'Brien, Oberbauer and Clark [33] have explored variance in tropical tree sap flux response to environmental factors among 10 different species and successfully developed a sap flux model that could account for over 70% of the observed variation, with species-specificity having a smaller effect than external factors such as tree crown position. In agreement with this, Meinzer, Goldstein and Andrade [32] found a convergence of plant regulation of water transport among 23 tropical species that illustrated a strong correlation between sap flux density and DBH. Similarly, an analysis of sap flux density for seven tropical species spanning heights between 10 and 30 m found tree height to be highly related to daily tree water use, with shorter trees taking up less water than taller trees [34].

Our findings conflict with the aforementioned work where water use was highly correlated with height and DBH. For our pre-tree death results, a possible explanation for this could be attributed to both a low sample size and the possibility of the opportunistic placement of one of the three understory trees used within this study. As one of our understory trees, *Trophis mexicana*, previously had an average sap flux of 19.98 kg m$^{-2}$ h$^{-1}$ (a sap flux much higher than the other two understory trees combined, as well as higher than any upper canopy tree pre-tree death), it is reasonable to speculate that this one tree may have skewed these measurements (Figure 3c). As for our post-tree death results, it appears that understory measurements were also skewed but not by the same sample tree. Where *T. mexicana* saw a decrease from 19.98 kg m$^{-2}$ h$^{-1}$ to 11.87 kg m$^{-2}$ h$^{-1}$, another understory tree, *Cupania macrophylla*, saw an increase of water usage from 11.61 kg m$^{-2}$ h$^{-1}$ to 25.13 kg m$^{-2}$ h$^{-1}$ (Figure 3c). While a drastic reshuffling of water use superiority seemed to only have occurred in understory species, this event provides nominal evidence for the effect of spatial heterogeneity as well as possible species-specific water use differences in our study.

As mentioned previously, treefall gaps and their associated effects are presumed to be a driver of maintaining species diversity in tropical forests [35]. However, these disturbances vary greatly between cases, and massive treefall gaps caused by the death of emergent species, such as *M. anisophyllum,* may increase as drivers of tree mortality become more frequent [36]. Nonetheless, this study highlights how tree water use dynamics can be altered under such death. Despite this work only representing a snapshot in time of the current state of tree water use within our forest plot, it is feasible to suggest that increased emergent tree death promoted by changing climate or accelerated deforestation could be an increasingly significant source of species turnover and community assembly in the future.

It is also important to note that while the death of an emergent tree promotes ecological succession of understory, opportunistic tree species, the ecological role that emergent trees play within tropical forests cannot be understated. While emergent trees play a vital role in carbon sequestration and water cycling [31], they also disproportionately account for a significant amount of aboveground biomass within tropical forests [37–40]. These massive pools of biomass not only serve a role at the global biogeochemical cycle but also locally serve as necessary habitat that sustains flora and fauna diversity within tropical forests, allowing for the retention of their structural complexity [37,41]. Fragmentation and hydraulic stress are able to potentially increase mortality rates of these traditionally robust and long-living trees [41]. Because of this, it will become increasingly important to further understand how these species locally influence forest structure in the context of prioritizing land for preservation and general reforestation efforts within these environments, especially under a changing climate where these stressors may become exacerbated.

Here, we highlight the need to better understand how water and light resources may be redistributed in the face of emergent species death, especially due to the unknown effects of climate change on tropical forests and the disproportionate amount of aboveground biomass and water recycling that emergent trees are responsible for within these ecosystems. Future studies assessing treefall gap dynamics in tropical forests should invoke

similar analysis on a broader scale along differing climate and moisture gradients, using a higher number of species and individuals, and while consistently monitoring how forest regeneration alters light regimes. This, done in conjunction with sampling across various plant functional groups, could potentially allow for researchers to better distinguish if water use dynamics are more constrained to either canopy layer or by life history traits and how the relationship between these two variables may change as canopy closure develops.

## 5. Conclusions

By assessing the sap flux of understory and overstory trees, this study highlights how relative water use may be redistributed among all canopy levels, as well as providing support for the hypothesis that understory trees may have a greater ability to take advantage of the release of light, drier leaves, and additional water resources than upper canopy species. It is unclear whether the observations made in this study were primarily driven by the increase of canopy penetrating light caused by the death of the emergent *M. anisophyllum* tree or due to year-to-year differences in micrometeorological conditions. However, the results shown here are more likely indicative of light/forest structure effects, due to greater percentage increase in sap flux within understory species and higher gap fraction and PAR reaching the lower canopy. While this study only represents a single canopy gap formation event, the significant impact of this one emergent tree death on the surrounding forest structure and PAR highlights the necessity of a deeper understanding of the impact of treefall-generated forest gaps and the role that they may play in species turnover and community assembly in the years to come.

**Author Contributions:** M.R.F.III conducted 2019 sap flux measurements, the subsequent analysis, and wrote the manuscript. L.M.T.A. conducted 2015 sap flux measurements and processed them. G.R.M. provided the MET tower and canopy light penetration data. G.W.M. provided the original research idea for this study, directed both 2015 and 2019 sap flux sensor installations, and served as lead PI for the associated NSF project. All authors have read and agreed to the published version of the manuscript.

**Funding:** This research was funded by the Texas A&M MSC L.T. Jordan Institute Fellows program and the LAUNCH International Research Grant, as well as the Texas A&M Department of Ecology and Conservation Biology. This project was conducted as an extension of a National Science Foundation Research Experiences for Undergraduates program (Award No. EAR-1659848) as well as being funded by the U.S. Department of Energy, Office of Science, Biological and Environmental Research (Grant DE-SC0010654).

**Institutional Review Board Statement:** Not applicable.

**Informed Consent Statement:** Not applicable.

**Data Availability Statement:** Data available upon request sent to the corresponding author.

**Acknowledgments:** We would like to acknowledge the Texas A&M Ecohydrology of Tropical Montane Forests NSF-REU program as well as the mentors for continued support of this research. we would also like to acknowledge the 2019 Moore Lab group for assistance in data analysis and sensor construction. Special thanks to the Soltis Center staff for their support.

**Conflicts of Interest:** The authors declare no conflict of interest. The funders had no role in the design of the study; in the collection, analyses, or interpretation of data; in the writing of the manuscript; or in the decision to publish the results.

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
