# Peer review of "Assessing Forest Level Response to the Death of a Dominant Tree within a Premontane Tropical Rainforest"

_forests, doi:10.3390/f12081041_

Round 1
Reviewer 1 Report
This study measured sap flux of overstory and understory trees after falling of an emergent tree via thermal dissipation probes over a year in a premontane tropical rainforest site, and compared water use of these trees with values derived before tree falling from a published paper to assess potential tree water use changes among these trees with different canopy positions. In general, the study was well designed and conducted. The manuscript is concise and clear, and easy to follow. Results suggested that the tree falling increased the stand openness and light penetration but did not significantly alter stand micrometeorology. And there was an increase of sap flux density in understory trees after the emergent tree falling comparing with overstory trees. Even though this increase was not statistically significant, I think it is still quite interesting to see a much larger increase in those understory trees comparing with nearly no increase in the overstory. It is evidence of resource (water) redistribution after the dominant tree falling. Overall, the study provides useful information on the effects of the emergent tree falling on forest resource (water) utilization dynamics at a small scale and is a valuable addition to the literature.
Suggestions and comments:
- L23, some numerical results regarding these significant changes would be nice. Also consider listing increase (numerical results, e.g., the percentage increased) of sap flux density, particularly in those understory trees, even the increase was insignificant.
- L86, ‘as well as championed as the highest water user in the world according to SAPFLUXNET [17,18].’. You may consider citing the SAPFLUXNET data paper for this sentence. It is published now.
Poyatos, R., Granda, V., Flo, V., Adams, M. A., Adorján, B., Aguadé, D., ... & van der Tol, C. (2021). Global transpiration data from sap flow measurements: the SAPFLUXNET database. Earth System Science Data, 13(6), 2607-2649.
- L131, I have a few questions regarding the TDP method in general. How did you account for, a) wound effect, b) radial changes/profile of sap flux density in those large trees, and c) details about the determination of delta Tm (zero flow condition). Please include some brief discussion if you did not account for these errors.
- L135-136 ‘levels with suppressed trees receiving one sensor per tree while both’. I guess one sensor for suppressed trees is under consideration of their relatively small DBH? Would be good to give an argument here to avoid potential confusion. Also, the direction of two sensors on large trees? I guess North and South? Please provide.
- L149, were the same tree measured in Aparecido et al 2016? Please provide a brief description of their measurement (tree characteristics, number of sensors, which method, TDP? etc.), or simply say the same setup was used in Aparecido et al.
- L166, please define extreme outlier, outside of 2*SD?
- L189-190, please provide height for temperature and RH measurements.
- L230-231, please provide reasons for data omission? Sensor failure?
- L237, what numbers after +- stand for? I saw you indicate it is SD in the figure caption, but it would be good to also describe it in the text when you first give it.
- L254, it would be good if you could somehow indicate the canopy level of these species in the fig3c itself, maybe group them according to their canopy position and add additional labels below the species name to indicate their canopy levels.
- L258, I would not call these ‘climate data’ as climate indicates long-term weather patterns, meteorological or even micrometeorological data sounds more appropriate. Also, I think logically you should show meteorological data first as these data indicate that environmental conditions at least the background annual weather pattern (rainfall amount, temperature, etc.) is comparable between selected years, no extreme weather pattern ever happened in one of the years (e.g., extreme drought), only then your water use comparisons are convincing, and differences observed would be results of the tree falling instead of confounding factors (e.g., interannual weather pattern).
- L271, I found the line color/style shown in fig4a is somehow confusing, please consider revising. Maybe same color for PAR at the same height but different line style for different years (e.g., purple for PAR at 10 m and dashed line for 2017 and solid line for 2019?).
Author Response
This study measured sap flux of overstory and understory trees after falling of an emergent tree via thermal dissipation probes over a year in a premontane tropical rainforest site, and compared water use of these trees with values derived before tree falling from a published paper to assess potential tree water use changes among these trees with different canopy positions. In general, the study was well designed and conducted. The manuscript is concise and clear, and easy to follow. Results suggested that the tree falling increased the stand openness and light penetration but did not significantly alter stand micrometeorology. And there was an increase of sap flux density in understory trees after the emergent tree falling comparing with overstory trees. Even though this increase was not statistically significant, I think it is still quite interesting to see a much larger increase in those understory trees comparing with nearly no increase in the overstory. It is evidence of resource (water) redistribution after the dominant tree falling. Overall, the study provides useful information on the effects of the emergent tree falling on forest resource (water) utilization dynamics at a small scale and is a valuable addition to the literature.
- L23, some numerical results regarding these significant changes would be nice. Also consider listing increase (numerical results, e.g., the percentage increased) of sap flux density, particularly in those understory trees, even the increase was insignificant.
AUTHOR RESPONSE: We appreciate the feedback and agree with the reviewer. We have re-worded portions of the abstract and added the percentage differences seen for sap flux density, light, and gap fraction before and after the tree fell. The revised portion reads: “…Hemispherical photography indicated a 40% increase in gap fraction as a result of changes in canopy structure after the treefall gap. Micrometeorological differences (e.g. air temperature, relative humidity and vapor pressure deficit (VPD)) could not explain the observed trends. Rather, light penetration, as measured by sensors within the canopy, increased significantly in 2019. One year after the tree fell, the water usage of trees across all canopy levels increased modestly (15%). Moreover, average water usage by understory trees increased by 36%, possibly as a result of the treefall gap, exceeding even that of overstory trees.” (L22-29).
- L86, ‘as well as championed as the highest water user in the world according to SAPFLUXNET [17,18].’. You may consider citing the SAPFLUXNET data paper for this sentence. It is published now.
Poyatos, R., Granda, V., Flo, V., Adams, M. A., Adorján, B., Aguadé, D., ... & van der Tol, C. (2021). Global transpiration data from sap flow measurements: the SAPFLUXNET database. Earth System Science Data, 13(6), 2607-2649
AUTHOR RESPONSE: Thank you! We have added the citation to L85.
- L131, I have a few questions regarding the TDP method in general. How did you account for, a) wound effect, b) radial changes/profile of sap flux density in those large trees, and c) details about the determination of delta Tm (zero flow condition). Please include some brief discussion if you did not account for these errors.
AUTHOR RESPONSE: We followed the same sap flux measurement and data processing protocols detailed by Aparecido et al. (2016) for a comparable dataset and consistency in methodology. As discussed in the aforementioned paper, zero flow was considered whenever VPD reached or close to 0 kPa. The mV or Js at that specific moment was set as TM in the Granier equation. Considering that the study site has high moisture levels, especially in the wet season, that level was reached daily overnight, which was confirmed through the instrumentation in the canopy tower. In regards to wound effects, due to the constant maintenance required on the sap flux network, sensors were exchanged often in 2015 and re-installed for 2019 comparisons. When reinstalled, the sensors were placed close to the original position, but further enough to not be affected by the wounded location. Thus, we do not believe that any of the sensor data shown in this manuscript were affected by sensor wounding. Finally, sap flux density rates were not corrected for radial changes since all the trees sampled had sapwood area deeper than 2 cm. Additionally, a side-project using longer TD probes showed that all the trees in the plot have highly active sapwood area beyond the 2 cm depth. To account for all these important factors in TDP methodology, a portion of “The collection of 2019 (post-tree death) water use measurements followed the same protocols used in Aparecido et al. 2016 [17] and consisted of continuous diurnal measurements of sap-flux density (Js, kg m-2 h-1) using 12 heat dissipation sensors [22] across 3 suppressed, 4 mid-story, and 2 subdominant trees (Table 1). The number of sensors varied by tree size with suppressed trees receiving one sensor per tree while both mid-story and subdominant trees received two sensors when possible. However, due to sensor failure and time constraints, not all subdominant and mid-story trees received multiple sensors. For trees with multiple sensors, the mean value for both sensors was used as the tree’s Js. Sensors were constructed as detailed in Phillips, et al. [23] and installed in the outermost xylem of our sample trees over a 5 week period between the months of June and July in 2019. All sensors were installed at breast height (1.30 m) or above tree buttresses, if pre-sent, and each set of sensors contained probes 2 cm in length (as recommended by Clearwater, et al. [24]). 2015 sensors were preferentially placed on side slopes when acces-sible and 2019 sensors were installed in areas clear of previous sensor wounds and placed opportunistically due to terrain. Wounding effects were minimized by new installations in 2019. Following Aparecido, Miller, Cahill and Moore [17], radial changes in sap-flux density were not taken into account.” (L130-146)
- L135-136 ‘levels with suppressed trees receiving one sensor per tree while both’. I guess one sensor for suppressed trees is under consideration of their relatively small DBH? Would be good to give an argument here to avoid potential confusion. Also, the direction of two sensors on large trees? I guess North and South? Please provide.
AUTHOR RESPONSE: Indeed, suppressed trees had smaller DBH, and therefore had only one sensor installed on each stem. When the DBH of sub-dominant and midstory trees were also small, but their canopies were taller than suppressed trees, they received one sensor; and when DBH was larger, two or even three sensors were installed (which were not included in this dataset). Due to the terrain, aspects such as the direction in which sensors were installed were not systematic but more opportunistic in terms of where I could actually install sensors, however, all sensors were installed above buttresses or at DBH as much as possible, as well as, away from locations were sensors had been previously installed. For better clarification in the manuscript, we have added the following to the text (already shown in the previous comment): “The number of sensors varied by tree size with suppressed trees receiving one sensor per tree while both mid-story and subdominant trees received two sensors when possible. However, due to sensor failure and time constraints, not all subdominant and mid-story trees received multiple sensors.” (L133-136)
- L149, were the same tree measured in Aparecido et al 2016? Please provide a brief description of their measurement (tree characteristics, number of sensors, which method, TDP? etc.), or simply say the same setup was used in Aparecido et al. (2016)
AUTHOR RESPONSE: The trees used in this study are a subset of the trees measured by Aparecido et al. (2016), and this is now specified in L151-152. Tree characteristics are shown in table 1, and we have added ‘number of sensors’ to the table.
- L166, please define extreme outlier, outside of 2*SD?
AUTHOR RESPONSE: We have rewritten this section and it no longer refers to ‘outliers’ specifically (L 167-169: “The data were screened for erroneous values due to sensor failure or out-of-range voltage readings, which were indicative of power loss, insect damage, or other environmental factors.”). The 2019 sap flux data were not proofed based on the 2*SD method. Instead, we identified unrealistic spikes in the data, which were later filtered out using a linear regression to extrapolate an estimated sap flux rate.
- L189-190, please provide height for temperature and RH measurements.
AUTHOR RESPONSE: The height of the temperature and RH probes are now addressed in L192.
- L230-231, please provide reasons for data omission? Sensor failure?
AUTHOR RESPONSE: The data omission and filtering were indeed due to sensor failure and. Upon further consideration of how the original 15 day data omission may have been biasing our original estimates, we also changed our analysis so that only weather data collected during the sap-flux sampling period was compared and used daily totals instead of monthly totals for these comparisons between years. Furthermore, for PAR data we compared data across the entire year of 2017 and 2015 where possible in order to better analyze differences in these parameters before and after tree-gap formation. We have now re-written this section as: “Daily averages of RH, T, and VPD MET tower data, spanning the same days as our sap-flux measurements, were compared as yearly averages using a two-tailed two sample t-test assuming equal variance. Days where all or most data were missing for either year were omitted from analysis (n > 95 days for all MET parameters). Annual rain gauge data were analyzed similarly, however no data filtering was necessary and monthly measurements were compared instead of daily averages (n = 6). Additionally, daily averages of vertical PAR profiles were also analyzed through linear regression analysis in R [25] using the ‘car’ package [26], and averages of the entire year were analyzed ( n > 250 days for each sensor height). This was done in conjunction with a two-tailed two sample t-test assuming equal variance in order to better observe the effect of year (pre- and post-tree death), sensor height, and the interaction between year and sensor height (i.e. how the effect of sensor height may be different pre- and post-tree death) under a type 3 sums of squares analysis (regression = logPAR ~ Year + Height + (Year × Height)). PAR data were log-transformed and sensor height was interpreted as integers rounded to the nearest meter prior to regression analysis. Lastly, November - December of 2019 PAR data were nearly completely missing for all sensor heights in 2019 (Figure 3a), resulting in no daily com-parisons being done for data within these months for either year.” L223-239.
- L237, what numbers after +- stand for? I saw you indicate it is SD in the figure caption, but it would be good to also describe it in the text when you first give it.
AUTHOR RESPONSE: Thank you for the pointing this out. We have now specified on L242 that ± refers to standard error, which is different than the standard deviation (SD) shown in what is now figure 4c.
- L254, it would be good if you could somehow indicate the canopy level of these species in the fig3c itself, maybe group them according to their canopy position and add additional labels below the species name to indicate their canopy levels.
AUTHOR RESPONSE: Great feedback! We have now added tree size subcategories to what is now Figure 4C.
- L258, I would not call these ‘climate data’ as climate indicates long-term weather patterns, meteorological or even micrometeorological data sounds more appropriate. Also, I think logically you should show meteorological data first as these data indicate that environmental conditions at least the background annual weather pattern (rainfall amount, temperature, etc.) is comparable between selected years, no extreme weather pattern ever happened in one of the years (e.g., extreme drought), only then your water use comparisons are convincing, and differences observed would be results of the tree falling instead of confounding factors (e.g., interannual weather pattern).
AUTHOR RESPONSE: We agree with the reviewer’s comment. We have now substituted the word “climate” throughout the text, when appropriate, with either “weather” or “micrometeorological”. We have also re-arranged the results section to show weather data first (L240).
- L271, I found the line color/style shown in fig4a is somehow confusing, please consider revising. Maybe same color for PAR at the same height but different line style for different years (e.g., purple for PAR at 10 m and dashed line for 2017 and solid line for 2019?).
AUTHOR RESPONSE: We appreciate and agree with your comment. With recommendations from both reviewers, we now have the years differentiated by color and the different sensor heights differentiated by shapes as denoted in the legend of this figure. Furthermore, based on other recommendations, this figure is now 3a instead of 4a.
AUTHOR CLARIFICATION ON WEATHER DATA:
As mentioned in response to comment 8 and written in L223-239 , weather data was re-analyzed to be able to assess weather during the same time period that sap-flux comparisons were made. This has resulted in significant differences being found for all weather parameters except for precipitation. Nonetheless, the fact that these relative humidity and air temperature parameters differed on average less than 10% between years, leads us to reason that changes in forest structure and PAR are still the main factors influencing our measurements. Especially when considering that PAR increased 40% and was measured within the our forest plot as opposed to just outside the forest (as was the case for our weather sensors). These comparisons have now been updated in the results, discussion, and conclusion section of the manuscript. Again, thank you very much for your insightful and kind feedback!
Reviewer 2 Report
I love the concept of the study. It is unique. The manuscript is well-written, and the authors tried to discuss the mechanisms exhaustively behind this canopy disturbance. However, I also have some concerns regarding the analysis. The measurements were taken in July-Dec, which is a wet season. In the tropics, the wet season has high rainfall amount that could saturate the trees. It would have been a lot different when the measurement was also done during the dry season. A study in an Amazonian forest on water use efficiency highlights these contrasts, and maybe you can see some insights in these papers:
https://www.mdpi.com/1999-4907/10/1/14
https://nph.onlinelibrary.wiley.com/doi/10.1111/j.1469-8137.2010.03245.x
I would say the study is still worth publishing despite some flaws if authors may make a little more effort to dig deeper into the wet season water usage and climate relationships (e.g. PAR, temp, rainfall, VPD etc) or biological relationships (e.g. biomass, LAI, etc). These climate and biological variables must also be wet season only. You can even remove the effect of rainfall to see the normalized sapflux variability alone. Light is the major controlling factor in ecosystem productivity during the wet season. Productivity and water usage are coupled. Hence, maybe the authors can explore along this line. Besides, linear regression analysis was mentioned in the methods, but I could not see any analysis results in figures or text.
In the context of no significant differences between pre-and-post disturbance in the study, I would say its because the gap may not be big enough to cause significant effect. Given the diversity in species composition and structural characteristic of the site, disturbances such as small as this may not that be so impactful unless exacerbated by major extreme events such as drought. Still, I commend the authors for coming up with this unique idea not many researchers have done, and it is just a first step moving forward, thus deserve publication for the scientific community to know.
|
Line/s |
Comments |
|
Fig 2 |
The figure captioned seemed hanging at the end. |
|
142 |
(.) after 2017 was misplaced |
|
168-169 |
What is the percentage of gaps for the entire measurement period? |
|
Fig 3 |
Maybe it is worthy to provide the significant differences in the figures represented by different letters or asterisks |
|
Fig 4 |
Panel A on PAR, maybe it would be nicer to only use one color for pre-death and another color for post-death and use symbols only for different years so as to easily see pre-and post death differences in PAR |
|
285 |
This can be more meaningful if biomass change between pre-and-post disturbance was presented |
|
291 |
But your result has no significant differences in sapflux |
|
298-31 |
You can have explored this sapflux-climate or sapflux-tree growth relationships on this regard to back up your claims |
|
318-321 |
You could have removed this outlier |
Author Response
I love the concept of the study. It is unique. The manuscript is well-written, and the authors tried to discuss the mechanisms exhaustively behind this canopy disturbance. However, I also have some concerns regarding the analysis. The measurements were taken in July-Dec, which is a wet season. In the tropics, the wet season has high rainfall amount that could saturate the trees. It would have been a lot different when the measurement was also done during the dry season. A study in an Amazonian forest on water use efficiency highlights these contrasts, and maybe you can see some insights in these papers:
https://www.mdpi.com/1999-4907/10/1/14
https://nph.onlinelibrary.wiley.com/doi/10.1111/j.1469-8137.2010.03245.x
AUTHOR RESPONSE: Thank you for the supportive words and feedback towards our study and manuscript! Indeed, we concentrated our study in the wet season mostly due to logistical reasons. Firstly, the tree fell late 2019, and due to the 2020 pandemic, the group was not able to return to the site and maintain the sap flow sensors that would have gathered the data for the dry season (Feb-May). Secondly, the manuscript presented is part of the undergraduate thesis of the lead author, Manuel Flores, of which was completed Spring/2020. Therefore, the data shown here was constrained to those wet season months and then compared to the partial time period and partial trees from Aparecido et al. (2016).
We do recognize that the dry season would have provided additional insights into forest stand dynamics after large tree fall. However, we believe that investigating tree water use of various canopy levels during the rainy season shows that even with the added stressors of canopy wetness, thicker canopy boundary layers, and occasional lower light intensity, the gap provided favorable conditions for the understory canopy layers to dry and access light. Based on this reflection, in addition to your feedback, we have added and rewritten L281-293 of the Discussion, which reads: “We hypothesized that a single treefall gap results in more light penetrating deeper into the canopy, which is expected to increase photosynthesis and air circulation that dries out trees and promotes higher water usage of the understory plants. Although we found high tree-to-tree variability for the 9 individuals analyzed in this study and no statistically significant response in tree water use between pre- and post-emergent tree death, average understory tree water use trended strongly upward after the gap formed, surpassing that of overstory trees. Light gap fraction was clearly higher and while this measurement may have also been affected by other treefalls in the area, PAR penetrated lower in the canopy after the tree fell. The effect may have been diminished during the wet season when the majority of our data were collected. More sunlight in the drier months could have enhanced water use by understory trees. On the contrary, more air circulation in the canopy gap to dry out wet leaves in the wet season could be a critical driver of observed differences.”
I would say the study is still worth publishing despite some flaws if authors may make a little more effort to dig deeper into the wet season water usage and climate relationships (e.g. PAR, temp, rainfall, VPD etc) or biological relationships (e.g. biomass, LAI, etc). These climate and biological variables must also be wet season only. You can even remove the effect of rainfall to see the normalized sapflux variability alone. Light is the major controlling factor in ecosystem productivity during the wet season. Productivity and water usage are coupled. Hence, maybe the authors can explore along this line.
AUTHOR RESPONSE: Thank you for this in depth feedback! A multiple regression utilizing many of the parameters mentioned in this study would have been best, however, due to the amount of data missing from our 2015 weather data, and the fact that our PAR data is from 2017 and not 2015, we are hesitant to conduct any analysis beyond a simple estimate of “before” and “after” analysis because of these constraints. Nonetheless, we did run a regression analysis concerning our PAR data to assess the effects of sensor height as well as year in order to analyze how this tree disturbance may have altered light intensity and differing canopy levels (L242-246).
Besides, linear regression analysis was mentioned in the methods, but I could not see any analysis results in figures or text.
AUTHOR RESPONSE: We apologize for the lack of clarity, and agree that this sentence was confusing. The linear regression mentioned on L235 refers to the relationship between PAR rates across sensor height and year, and the results from this analysis is written on L242-246 in the Results section. For better clarity, we have rewritten the regression in our methods section as “ regression = logPAR ~ Year + Height + (Year × Height)” on (L235). Results for the regression mentioned in the methods can now be found on L242-246 as “We found significant differences between years in PAR (2017 = 53.25 ± 1.49 (SE) μmol m-2 s-1 , 2019 = 74.60 ± 1.85 μmol m-2 s-1 , P < 0.0001), and significant effects of year and light sensor height within our linear regression analysis (PYear < 0.0001and PHeight < 0.0001). However, there was no significant effect for the interaction term Year x Sensor Height for PAR (P = 0.63, Figure 3c)”.
In the context of no significant differences between pre-and-post disturbance in the study, I would say its because the gap may not be big enough to cause significant effect. Given the diversity in species composition and structural characteristic of the site, disturbances such as small as this may not that be so impactful unless exacerbated by major extreme events such as drought. Still, I commend the authors for coming up with this unique idea not many researchers have done, and it is just a first step moving forward, thus deserve publication for the scientific community to know.
AUTHOR RESPONSE: Again, thank you for your supportive words and feedback. We agree that the size of the gap is a major factor in the water use and microclimatic dynamics seen. However, we are not able to quantify the actual size of the gap, as well as compare it with other studies that analyze the effects of different gap sizes on forest productivity. In light of this limitation, we have added “gap size influence” across the Discussion. But, more specifically, we have added the following sentence: “The size of the treefall gap may have also been a factor that we could not account for in this study. Small but significant differences in weather between years could possibly ex-plain the differences observed, especially when considering the higher VPD means pre-sent in 2019 measurements. Nonetheless, the overstory had previously been responsible for ~22% more water consumption than understory individuals pre-tree death; the shift to overstory trees using ~2% less water than understory trees suggests a growth response from the forest understory after the gap was formed.”(L293-299).
|
Line/s |
Comments |
|
Fig 2 |
The figure captioned seemed hanging at the end. |
|
142 |
(.) after 2017 was misplaced |
|
168-169 |
What is the percentage of gaps for the entire measurement period? |
|
Fig 3 |
Maybe it is worthy to provide the significant differences in the figures represented by different letters or asterisks AUTHOR RESPONSE: We appreciate the feedback! We have now added asterisks to what is now figure 4b. |
|
Fig 4 |
Panel A on PAR, maybe it would be nicer to only use one color for pre-death and another color for post-death and use symbols only for different years so as to easily see pre-and post death differences in PAR |
|
285 |
This can be more meaningful if biomass change between pre-and-post disturbance was presented |
|
291 |
But your result has no significant differences in sapflux. AUTHOR RESPONSE: This sentence has now been reworded to make this more apparent, and it reads “Our study found no evidence that the gap induced more stressful conditions where the emergent canopy partially shaded and protected overstory canopies” (L300-301). Thank you! |
|
298-31 |
You can have explored this sapflux-climate or sapflux-tree growth relationships on this regard to back up your claims AUTHOR RESPONSE: Thank you for this additional suggestion. Unfortunately, due to the weather data constraints and lack of post-tree death biomass estimate, we are unable to explore these factors robustly with the data that we have concerning this site. This will be improved for any further analysis moving forward. Again, thank you! |
|
318-321 |
You could have removed this outlier AUTHOR RESPONSE: We agree that removing the outlier would have made the trends more apparent and statistically significant. However, due to the already small sample size of overlapping trees between this study and Aparecido et al., (2016), we decided to keep the tree with different responses in the datasets. Additionally, we cannot complete dismiss the fact that these contradictory trends were possibly due to species-specific traits, as well as their location within the stand. Further investigation is needed to understand why certain species respond differently to inter-canopy microclimatic variation. |
AUTHOR CLARIFICATION ON WEATHER DATA:
Upon further consideration of how our original 15 day data omission may have been biasing our original estimates, we changed our analysis so that only weather data collected during the sap-flux sampling period was compared and used daily totals instead of monthly totals for these comparisons between years. Furthermore, for PAR data we compared data across the entire year of 2017 and 2015 where possible in order to better analyze differences in these parameters before and after tree-gap formation. We have now re-written this section as: “ Daily averages of RH, T, and VPD MET tower data, spanning the same days as our sap-flux measurements, were compared as yearly averages using a two-tailed two sample t-test assuming equal variance. Days where all or most data were missing for either year were omitted from analysis (n > 95 days for all MET parameters). Annual rain gauge data were analyzed similarly, however no data filtering was necessary and monthly measurements were compared instead of daily averages (n = 6). Additionally, daily averages of vertical PAR profiles were also analyzed through linear regression analysis in R [25] using the ‘car’ package [26], and averages of the entire year were analyzed ( n > 250 days for each sensor height). This was done in conjunction with a two-tailed two sample t-test assuming equal variance in order to better observe the effect of year (pre- and post-tree death), sensor height, and the interaction between year and sensor height (i.e. how the effect of sensor height may be different pre- and post-tree death) under a type 3 sums of squares analysis (regression = logPAR ~ Year + Height + (Year × Height)). PAR data were log-transformed and sensor height was interpreted as integers rounded to the nearest meter prior to regression analysis. Lastly, November - December of 2019 PAR data were nearly completely missing for all sensor heights in 2019 (Figure 3a), resulting in no daily com-parisons being done for data within these months for either year.” L223-239.
This has resulted in significant differences being found for all weather parameters except for precipitation. Nonetheless, the fact that these relative humidity and air temperature parameters differed on average less than 10% between years, leads us to reason that changes in forest structure and PAR are still the main factors influencing our measurements. Especially when considering that PAR increased 40% and was measured within the our forest plot as opposed to just outside the forest (as was the case for our weather sensors). These comparisons have now been updated in the results, discussion, and conclusion section of the manuscript. Again, thank you very much for your insightful and kind feedback!
Round 2
Reviewer 1 Report
I appreciate the author's revision, all my suggestions are well addressed!
Reviewer 2 Report
Thank you for addressing my concerns. I have no further comments.